# Synthesis of Water-Soluble Copolymers of *N*-vinylpyrrolidone with *N*-vinyldithiocarbamate as Multidentate Polymeric Chelation Systems and Their Complexes with Indium and Gallium

**DOI:** 10.3390/molecules25204681

**Published:** 2020-10-14

**Authors:** Nikolay I. Gorshkov, Andrey Yu. Murko, Irina I. Gavrilova, Marina A. Bezrukova, Albert I. Kipper, Valerii D. Krasikov, Evgenii F. Panarin

**Affiliations:** Federal State Budgetary Institution of Science Institute of Macromolecular Compounds, Russian Academy of Sciences (IMC RAS), 199004 Saint Petersburg, Russia; nitrobenzene@yandex.ru (A.Y.M.); lenchrom@hq.macro.ru (I.I.G.); bezrukova@imc.macro.ru (M.A.B.); kipper@imc.macro.ru (A.I.K.); krasikov@lenchrom.ru (V.D.K.); panarin@hq.macro.ru (E.F.P.)

**Keywords:** *N*-vinylpyrrolidone copolymers, metal–polymer complexes, dithiocarbamate, gallium, indium

## Abstract

Dithiocarbamate (DTC) derivatives of *N*-vinylpyrrolidone-*N*-vinylamine (VP–VA) copolymers were synthesized via reaction between the copolymers and carbon disulfide in alkaline medium; molecular masses of the products were 12 and 29 kDa; the VP:VDTC ratios were 94:6 and 83:17 mol.%. Complexation between the obtained DTC derivatives and metal ions (indium and gallium) was investigated. It was demonstrated that metal–DTC ligand complexes with 1:3 ratio between components were formed. Gallium metal–polymer complexes (MPC) were unstable in solution. Individual indium MPC were isolated and characterized by spectral and chromatographic methods. Unlike similar gallium MPC, they appeared to be stable in histidine challenge reaction.

## 1. Introduction

The chemistry of synthetic metal–polymer complexes (MPC) is an urgent and vast area of polymer science. MPC find wide application in various fields of science and engineering: analysis and fixation of heavy metal ions, design of biosensors, and electro-optical devices [1,2,3,4]. Besides, MPC demonstrate antimicrobial, antibacterial, antiviral, antifungal, and antitumor activities [5,6,7,8,9,10].

MPC are macromolecules containing metal ions, which, as a rule, are linked with polymer chains via chelate [11]. The use of metal–ligand interactions appears to be versatile in this context, since the interaction strength and dynamic nature of the complexes can, in principle, be tuned by variation of the metal ions, ligands, and linkers. Systems with various architectures can be formed, including crosslinked intramolecular structures [12,13] and intermolecular and self-assembling aggregates, in which metal ions play the role of anchor groups [13] or are localized on the surface of polymer coils [12]. Chelating units for metal binding can be distributed along polymer chains randomly, in blocks or microblocks; thus, a polymeric matrix can be considered as a complex polyfunctional chelating system. Properties of this complex system are controlled not only by characteristics of a polymeric carrier (molecular mass, structure, hydrodynamic parameters of polymer coil, the presence of ionogenic, hydrophobic, hydrophilic, or other functional groups), but also by the parameters of the spacer that separates a chelating unit from polymer backbone, steric parameters of chelate groups, their donor activity, and other factors [14,15]. Thus, the synthesis of MPC with predetermined properties is a challenging experimental task.

In this context, dithiocarbamates (DTC) (R,R’-N-S(S))seem to be promising compact and powerful bidentate ligand systems, which form strong complexes with the majority of metal ions [16,17]. High stability of the DTC chelating units is caused by the formation of a four-membered ring with delocalized negative charge [16,17]. DTC have long attracted the attention of researchers working in the field of MPC chemistry due to their compact sizes, which allow one to decrease their influence on the hydrodynamic and structural properties of macromolecules. DTC groups were successfully introduced into various hypercrosslinked macromolecular structures.

There have been a number of investigations on MPC, containing DTC chelators, mainly focused on the fixing of heavy metals from industrial wastewaters. Generally, DTC-containing polymers are insoluble in water and represent themselves as flakes, gels, nanoparticles, nanotubes, and grafted nanocomposites. The described DTC-containing polymeric matrices are based on polystyrenes, polyacrylamides, carbon nanotubes, etc. [18,19,20,21,22,23]. A wide range of DTC derivatives of carbohydrates (starch, chitosan) were prepared and investigated for this purpose [24,25,26].

Dithiocarbamate ligands are obtained by interaction between primary or secondary amines with carbon disulfide in the presence of a proton acceptor (sodium, potassium, or ammonium hydroxides) [17,27,28].

DTC and their complexes with metals find wide application in medicine for therapy of various diseases (for instance, [29,30]). DTC derivatives of chitosan find application in medicine as carriers for low molecular weight biologically active compounds [21,22,23,24,25,26,27,28,29,30,31].

In recent decades, researchers engaged in the development of novel radiopharmaceuticals have shown an increasing interest in introducing radionuclides (mainly metals) into various macromolecular objects (nanoparticles, micelles, dendrimers, flexible-chain water-soluble synthetic and half-synthetic polymers). Generally, radiolabeled polymers are intended for diagnostics and therapy of tumors [32,33,34]. These objects are promising materials for medicinal applications, since it is possible to realize their passive transport through cell membrane (enhanced and permeation retention (EPR) effect) [35], and to perform relatively easy modification of polymeric carriers by site-specific low molecular weight vectors, which have affinity for certain tissues. An example of the application of folic acid chitosan modified with dithiocarbamate units, labeled with^99m^Tc as potential agents for folatereceptor-mediated targeting is was described [36]. Thus, biologically compatible polymers decorated with DTC chelation units appeared to be stable in vitro and in vivo and are promising platform for development of novel macromolecular radiopharmaceuticals.

Flexible-chain polymers have a number of advantages as carriers of low molecular biological compounds (BAC) and metal radionuclides in contrast to rigid and semi-rigid nano particles and micelles [37]. Flexibility of the polymeric coil increases the probability of realization of an EPR effect in comparison with rigid macroaggregates, which have limited penetration and distribution within malignant cells and highly depend on size and shape. In this respect, poly-*N*-vinylpyrrolidone (PVP) and its copolymers as typical representatives of flexible-chain water-soluble polymers seem to be promising carriers for metal radionuclides. They possess biological activity, immunostimulating properties, act as carriers of low molecular weight biological BAC, and demonstrate weak complexing properties [38,39,40].

To obtain MPC based on *N*-vinylamide copolymers, chelating fragments are introduced into copolymers during polymerization (β-diketones, acrylic acid, crotonic aldehyde) [41,42,43] or grafted to functional groups present in macromolecules via polymer-analogous reactions [44,45] (iminodiacetic acid, Schiff bases). Copolymers of *N*-vinylpyrrolidone (VP) with allylamine (AA) [46] or vinylamine (VA) [47,48] are convenient for modification, since they contain reactive amino groups and can be relatively easily synthesized.

Radionuclides of three-valent metals are widely used in nuclear medicine as diagnostic (gallium-68) radiotracers for positive emission tomography (PET) imaging and indium-111 for tumor therapy due to optimal nuclear and physical properties [49].

It has been demonstrated previously [50,51,52] that VP–VA and VP–AA copolymers containing chelation units (1,4,7,10-tetraazacyclododecane-1,4,7,10-tetraacetic acid (DOTA), iminodiacetic acid (IDA)) have different complexating abilities toward indium and gallium ions. In the cases of VP–VA–DOTA and VP–VA–IDA copolymers, radiochemical yields with gallium-68 were up to 57–66% and 80%, respectively. For indium-111 and VP–VA–IDA copolymer, radiochemical yield was 98%. Thus, it could be assumed that compact low denticity ligand systems (IDA), attached to polymeric backbone, have advantages in comparison to bulky chelators, which essentially are oriented to the external surface of polymeric coil. Polymers containing low denticity chelation units can form intramolecular MPC, where metal ion acts as an anchor and have completely occupied the coordination environment [52,53,54].

Thereby, the goal of the present work was to synthesize water-soluble *N*-vinylpyrrolidone-*N*-vinylamine (VP–VA) copolymers, decorated with low denticity DTC chelation units, to prepare MPC based on these copolymer derivatives with three-valent metals (gallium and indium), and to estimate the stability of the obtained MPC in the media close to biological systems. Basic knowledge on the structure, composition, and stability of MPC is urgent for further development of radiolabeling procedures and evaluation of biological behavior.

## 2. Results and Discussion

### 2.1. Synthesis of VP–VDTCCo-Polymers

The synthesis of flexible-chain water-soluble VP-VDTC opolymers in aqueous media is a rather complicated task. Introducing DTC groups into macromolecules is well developed in the cases when polymers demonstrate low solubility in water or do not dissolve in it at all. As a rule, these reactions proceed under heterophase conditions or in strong aprotic solvents (DMFA) [18,19,20,21,22,23,24,25,26,27,28].

The initial VP–VA copolymers were synthesized according to the modified procedure described elsewhere [55]; molecular masses of the products were 12 × 10^3^ and 29 × 10^3^ Da; contents of amino groups were 6 and 18 mol.%. The copolymer of *N*-vinylpyrrolidone with *N*-vinylformamide was synthesized by free radical polymerization initiated by 2,2′-azobisisobutyronitrile (AIBN) in inert atmosphere (Scheme 1A, Section 3. Materials and Methods).

Then, formyl protecting groups were removed by boiling the product in the aqueous solution of 1 M hydrochloric acid (Scheme 1B). DTC groups were introduced in ethanol solution in the presence of a fourfold excess of potassium hydroxide; then, the solution was centrifuged to remove the precipitated potassium chloride (Scheme 1C). At the final stage, carbon disulfide (22-fold excess) was added. The reaction mixture was exposed at room temperature for 30 min at continuous stirring (Scheme 1D).

All synthetic procedures for VP–VDTC copolymers were carried out in inert atmosphere to prevent side oxidation reactions.

In order to remove potassium ethyl xanthogenate that was formed as a side product, VP–VDTC copolymer was precipitated from the reaction medium with ethyl acetate/acetone mixture, reprecipitated from ethanol solution into ethyl acetate, then dried under vacuum. The amount of introduced DTC groups was determined by elemental analysis (sulfur determination). Mass fractions of sulfur were 2.9% (for the copolymers containing 6 mol.% of DTC groups), and 8% (for the copolymers with 17 mol.% of DTC groups). This leads to the conclusion that the reaction between the copolymer and carbon disulfide proceeded in an almost quantitative yield. The prepared VP-VDTC copolymers were characterized by the following physicochemical methods: ^13^C NMR spectroscopy, UV spectroscopy, IR spectroscopy, high performance liquid chromatography (HPLC), and viscometry.

^13^C NMR spectroscopy turned out to be uninformative, since the signal corresponding to the resonance of the CSS group (about 209 ppm) is very broad and has low intensity. The same effect was observed for DTC derivatives of other polymers [55,56]. ^1^H NMR spectral data for synthesized copolymers are presented in the experimental section and show that observed resonances could be assigned to protons of polymeric backbone.

### 2.2. UV and IR Spectroscopy of VP–VDTC

UV spectra of VP–VDTC copolymers include absorption maximums typical of dithiocarbamates (at 255 nm (lgε~4.03), (π→π* electron transition in –C=S group and 287 nm (lgε~4.09) (n→σ* electron transition in C–S^−^ group) [17,18]). The appearing of corresponding bands in UV spectra indicatesattaching of the CSS groups to VP–VA copolymer chains.

In IR spectra of the isolated VP–VDTC copolymers, a number of vibration bands characteristic to the dithiocarbamate group: 1126 cm^−1^ (N–C–S(S)), 1043, 976, 951 cm^−1^ (thionic (C=S) group) (Figure 1) appeared in contrast to the starting copolymer. Intensive bands, which could be assigned to C=S asymmetric vibrations in the range 650–750 cm^−1^, are overlapped with polymer bonds.

### 2.3. Size-Exclusion (SEC) Chromatography of Co-Polymers

The obtained copolymers were studied by SEC in 0.2 M solution of sodium chloride, which is commonly used to suppress polyelectrolyte effects. Calibration dependence was plotted using the poly(*N*-vinylformamide) standards with narrow molecular mass distributions [57,58].

Since the VP–VA∙HCl and VP–DTC copolymers are strong polyelectrolytes, their retention volumes depend on the amount of charged groups and their polarities. For instance, in the case of copolymers with *M*_r_ of 12 and 29 × 10^3^ Da and content of NH_2_ groups equal to 18 mol.%, we observe a characteristic shift in chromatographic profile, and shapes of the peaks in chromatograms of VP–VA∙HCl and VP–VDTC copolymers change in comparison to that of the initial VP–VFA (Figure 2). In addition, the VP–VDTC peak shifts beyond the calibration dependence (Figure 2).

Retention volumes of the similar copolymers (VP–VFA and VP–VA∙HCl) containing 6 mol.% of polar groups coincided, and the retention volume of the VP–VDTC copolymer changed significantly (Figure 2 (3)). In cases of the VP–VDTC peak was considerably broadened and had a pronounced tail. Thus, due to the strong polyelectrolyte effect arising in solutions of VP–VDTC copolymers, a mixed sorption–exclusion mechanism is realized during chromatographic analysis. As a result, correct chromatographic determination of molecular masses and molecular mass distributions of samples appears difficult. Despite that, the noticeable shift of retention time and appearing of absorption at wavelength 254 nm indicate the formation of a novel polymer with attached CSS groups.

Thus, it could be concluded that stable water-soluble chain-flexible copolymers VP–VDTC of various compositions were firstly synthesized.

### 2.4. Synthesis and Investigation of Stoichiometry of M(Ga,In)–IDADTC (Bis (Carboxymethyl) Dithiocarbamate) Model Complex

The chemistry of metal dithiocarbamates is long and well studied within the framework of classical coordination theory. However, the coordination behavior of the ligands attached to polymer carriers is mainly determined by the properties of a macromolecule (its hydrodynamic and structural characteristics) and, to a lesser degree, by the properties of a chelating center.

Moreover, during complexation of metal ions with multiple DTC polymeric chelation ligand systems, some steric hindrances and spatial mismatches appear. These steric features prevent complete complexation between chelation groups and metal ions, which potentially results in formation of flocs with excess negative charges. Therefore, formation of inter- and intramolecular cross-linked structures can be expected in aqueous solutions upon interaction between metal ions with coordination number equal to 6 and the DTC-containing polymeric chelation system

The majority of metal complexes with dithiocarbamate ligands are insoluble in water, since they contain hydrophobic alkyl/aryl substituents, and, besides, in a number of cases, they have hypercrosslinked structure (for instance, see [21,22,23,24]), in which DTC groups are oriented inside a polymer coil.

Spectral data of metal DTC complexes in aqueous solutions are scarce and mainly relate to compounds soluble in organic solvents. Therefore, to model the coordination behavior of DTC ligands grafted to polymer chain and to compare the behavior of model complexes M(DTC)_3_ and that of the corresponding DTC, we synthesized a hydrophilic water-soluble DTC ligand, a derivative of IDA (potassium bis(carboxymethyl)dithiocarbamate, IDADTC).

The compound was prepared according to the modified procedure [59]. UV spectroscopy studies revealed that in water solution at pH 4, this ligand demonstrated absorption maximums at λ = 259 nm (π→π* transition) and at 287 nm (n→σ* transition). Upon complexation with indium ion, the first absorption maximum was shifted hypsochromically, and the magnitude of this shift increased during stepwise complexation. Thus, in the case of coordination between one metal ion and one ligand (M^3+^:L = 1:1), the peak was located at 251 nm, after 1:2 coordination, the band wavelength was 255 nm, and in the case of 1:3 coordination, this value was 257 nm (Figure 3). The intensity of the second absorption maximum (λ = 287 nm) decreased considerably. Thus, the obtained UV data on the stepwise complexation of water-soluble DTC ligand with metal ions allow one to estimate the coordination environment of metal ion in the polymeric DTC chelation system.

In the ^13^C NMR spectra of the complex with M:L ratio equal to 1:3, the following signals were observed: 202 ppm (CSS) (211 ppm: free ligand), 173 ppm (COO^−^), 61 ppm (–CH_2_^−^) (Figure 4). The ^1^H NMR spectrum contained a singlet signal at 4.38 ppm, down field shifted in contrast to free ligand (4.62 ppm), assigned to two equivalent methylene groups of the ligand. This structure of NMR spectra indicates coordination between the ligand and indium ion.

Thus, it can be concluded that coordination between the CSS chelate system and indium ion stabilizes this system due to the formation of a quasi-aromatic ring with delocalized negative charge.

Interaction between IDADTC and gallium ion at near neutral pH values did not lead to complexation. Moreover, the IDADTC ligand decomposed with the formation of iminodiacetic acid and carbon disulfide, as can be seen in Figure 5.

Apparently, these results are caused by the difference between the ionic radii of gallium and indium (0.62 and 0.80 Å, respectively) [60]; in the case of gallium ion, a strained and unstable Ga-SCS ring is formed.

A similar phenomenon was observed by the authors of [61] during HPLC separation of complexes with pyrrolidine- and diethyldithiocarbamates and a number of metal ions in water-organic media.

### 2.5. Synthesis and Investigation of Ga/In-VP–DTC MPC

MPC of indium with the VP–VDTC copolymers containing 94.6 mol.% of VP units and 6 mol.% of VDTC units, with molecular masses of 12 × 10^3^ and 29 × 10^3^ Da, were isolated from solution, purified by dialysis against water and freeze-dried. MPC were studied by IR spectroscopy and size exclusion chromatography; metal content was determined by inductively coupled plasma atomic absorption spectroscopy. Gallium MPC turned out to be stable only in solution and decomposed on attempted isolation.

Detailed studies of the manner of coordination between DTC chelation units introduced in a polymer backbone and metal ion, and estimation of stoichiometry of the formed complexes were performed using UV, IR, and ^1^H, ^13^C NMR spectroscopy.

#### 2.5.1. UV Measurements

Complexation of gallium and indium with polymer-based DTC chelating systems was studied by electronic absorption spectroscopy. The spectra of a series of solutions with constant concentration of chelating polymer (concentration of DTC groups *C*_VDTC_ = 5.25 × 10^−5^ mol/L) were taken. The M:L ratios and the corresponding concentrations of metal ions were varied from M:L = 1:1 (C_M_^3+^ = 5.25 × 10^−5^ mol/L) to M:L = 1:100 (C_M_^3+^ = 5.83 × 10^−3^ mol/L). Already at the M:L ratio of 1:1, the spectra contained one absorption maximum at 257 nm, and the band at 287 nm (typical of uncoordinated CSS group) disappeared. Thus, we can conclude that metal ions have hexacoordination surrounding. Further increases in metal concentration did not cause significant spectral changes (in the cases of both indium and gallium, Figure 6).

The above results indirectly indicate that behavior of a polymeric chelating system is totally different from that of low molecular weight analogs; here, complexating ability depends not only on donor activity of a chelation site, but even more so on conformation of a polymer carrier. The sterically strained coordination unit Ga-SSC-N is stabilized by polymer coil.

#### 2.5.2. IR Spectroscopy

IR spectra of indium MPC demonstrated a shift of the characteristic intensive band attributed to C=S valence vibrations from 960 to 976 cm^−1^; a number of less intense bands (N-C=S, 1126, 1088, 1051 cm^−1^) also shifted toward higher frequencies as compared to the spectrum of the initial polymer (Figure 7).

According to the data of inductively coupled plasma atomic absorption spectroscopy (iCAP), the content of indium in the copolymer was about 1.8 wt.% (i.e., 99% of metal ions were bound).

#### 2.5.3. SEC Measurements

The peaks in chromatographic profile of MPC between indium and VP–VDTC shifted toward lower retention volumes in comparison with chromatogram of the initial polymer; this indicates a decrease in hydrodynamic radii of the MPC caused by the formation of intramolecular crosslinks In(DTC)_3_ (Figure 8).

### 2.6. Molecular Hydrodynamic and Optic Investigation of Copolymers and In-VP–VDTC MPC

The following molecular parameters of the initial VP–VDTC copolymer and its MPC with indium were estimated: intrinsic viscosity [η] in water and 0.2 M NaCl, translational diffusion coefficient *D* and sedimentation velocity coefficient *S* in 0.2 M NaCl (24 °C), molecular masses *M*_sD_. Hydrodynamic radii (*R*_h_) were calculated from the dynamic light scattering (DLS) data; refractive index increments (*dn/dc*) were determined. The Tsvetkov–Klenin hydrodynamic constants were calculated by the formula *A*_0_ = (η_0_D/T) × ([η]·M_sD_)^1/3^ [62]. The results are presented in Table 1.

The copolymers with the composition of 82:18 mol.% did not form stable MPC with indium and decomposed in aqueous solutions in 24 h, which impeded correct measurements. Therefore, the copolymers with the composition of 94:6% were studied. These samples formed true solutions and were stable for prolonged periods of time. In the presence of indium ions, intrinsic viscosity [η] of macromolecules decreased, but the corresponding change in diffusion coefficient was not observed. Apparently, macromolecule asymmetry exerts a stronger influence on viscosity than on the translational friction coefficient. The presence of indium ions in polymeric complexes caused an increase in the refractive index increment. The trend has been toward increases in size (mass) homogeneity of the system as a result of complex formation (as compared to the initial polymer samples), which manifested itself in a uniquely determined sedimentation coefficient. The data from Table 1 show that *M*_r_ masses of the complexes do not differ significantly from *M*_r_ of the initial samples. The DLS data for indium MPC with higher *M*_r_ demonstrate that the hydrodynamic radius decreased in comparison with that of the initial copolymers, which indicates the formation of cross-linking MPC. Obtained MPC have hydrodynamic radii suitable for biological application, e.g., target transport of metal radionuclides with the realization of EPR effect.

It should be noted that monomeric units inside the polymeric chain in initial copolymers VP–VFA have statistic distribution [63]. Consequently, the statistic character of the VP–VDTC copolymerand irregular arrangement of M(DTC)_3_ fragments inside polymeric coil are also expected.

Thus, it can be inferred that VP–VDTC copolymers form MPC with indium ions with the maximum possible amount of ligands coordinated to the metal atom (M:L = 1:3).

A schematically proposed structure of the VP–VDTC MPC coordination environment of metal ion inside the VP–VDTC polymeric coil is depicted by Scheme 2. All possible coordination positions (6) in the metal coordination sphere are occupied with bidentate DTC ligands.

It should be emphasized that intramolecular structures are known in the chemistry of MPC and their formation depends on the architecture of polymers and synthetic conditions [12].

Thereby, stable and potentially suitable for biological application VP–VDTC–M^3+^MPC were synthesized and characterized.

### 2.7. Stability of MPC in Histidine Challenge Reaction (HCR)

An important characteristic of the obtained MPC is their stability in simulated biologically active media, namely, the histidine challenge reaction (HCR). Histidine is not only an important biological amino acid incorporated into blood proteins but serves as a strong chelating agent and forms complexes with most metal ions. HCR is a common test used to evaluate the stability of metal complexes or conjugates in the cases when it is difficult to estimate stability in vitro. In our case, the most convenient method for these investigations was UV–vis spectroscopy. UV spectra were registered in phosphate buffer at physiological pH (7.4). Figure 9 and Figure 10 present the data for the VP–VDTC copolymer and the corresponding MPC with *M*_r_ of 12 × 10^3^ Da and VP:VDTC ratio equal to 82:18 mol.%.

The solution of the initial copolymer was prepared immediately before measurements to avoid oxidation by air oxygen, then a stoichiometric amount of aqueous solution of metal salt (pH = 3.5) was added. UV spectra of MPC and MPC with 20-fold excess of histidine were taken after incubation for 2 h at 37 °C. It was demonstrated that indium MPC were stable in standard conditions (Figure 9), and gallium MPC underwent trans-chelation and decomposed to the initial copolymer and low molecular weight complex Ga(His)_2_^+^ (Figure 10).

The stability of gallium and indium MPC with VP–VDTC in HCR additionally confirms that the metal coordination sphere of metal ion is completely occupied with donor atoms of ligand.

## 3. Materials and Methods

### 3.1. Chemicals, Reagents and Materials

*N*-vinylpyrrolidone (*N*-VP, “Sigma-Aldrich”, St. Louis, MO, USA) and *N*-vinylformamide (*N*-VFA, “Sigma-Aldrich”, St. Louis, MO, USA) monomers, and 2,2′-azobisisobutyronitrile initiator (AIBN, Biolar, high purity grade, St.-Petersburg, Russian Federation) were used. The following solvents and reactants were used as received: *N*,*N*-dimethylformamide (“Vekton”, reagent grade, St.-Petersburg, Russian Federation), indium and gallium metals “Sigma-Aldrich”, St. Louis, MO, USA), concentrated hydrochloric acid (“Vekton”, analytical grade, St.-Petersburg, Russian Federation), carbon disulfide “Sigma-Aldrich”, St. Louis, MO, USA), acetone “Sigma-Aldrich”, St. Louis, MO, USA), isopropanol (“Vekton”, analytical grade, St.-Petersburg, Russian Federation), ethanol (94.5%, reagent grade, St.-Petersburg, Russian Federation), diethyl ether (“Vekton”, analytical grade, St.-Petersburg, Russian Federation), ethyl acetate (“Vekton”, analytical grade, St.-Petersburg, Russian Federation), potassium hydroxide (“Vekton”, analytical grade, St.-Petersburg, Russian Federation).

*N*-VP and *N*-VFA monomers were purified by distillation under vacuum (b.p. = 69 °C (3 mm Hg), n_D_^20^ = 1.5120; b.p. = 65 °C (4 mm Hg), n_D_^20^ = 1.4920, respectively). AIBN initiator was purified by recrystallization from ethanol:chloroform mixture (3:1), m.p. = 103 °C.

Phosphate-buffered saline (PBS) solutions were prepared by dissolving the preformulated tablets (Sigma-Aldrich) in 200 mL of Milli-Q water (mean resistivity > 18.2 Ω) to give [NaCl] = 0.138 M, [KCl] = 0.0027 M, and pH 7.4.

### 3.2. Synthesis of N-Vinylpyrrolidone Copolymers

Overall sheme of copolymer synthesis is presented on Scheme 2.

#### 3.2.1. Synthesis of *N*-Vinylpyrrolidone-*N*-VFA Copolymers

*N*-vinylpyrrolidone (2 g, 18 mmol), *N*-vinylformamide (0.32 g, 4.5 mmol), and AIBN (azobisisobutyronitrile)initiator (0.07 g, 0.43 mmol, 3 wt.% with respect to total monomer mass) were introduced into a heat-resistant glass ampoule. Isopropanol (55.6 mL) was added to the mixture. The ampoule was sealed in inert atmosphere and thermo stated at 65 °C for 24 h. When the reaction was complete, the mixture was evaporated using a rotary evaporator and precipitated into diethyl ether. The final product was dried under vacuum at 80 °C for 12 h. Compositions of the copolymers were determined from the ^1^H NMR data, namely, from the relationship between intensities of the signals attributed to the formamide NH group (8 ppm) and methylene (1.97 and 3.26 ppm) groups of the lactam ring. The copolymer yield was 1.8 g (77.6%).

^1^H NMR (500 MHz, D_2_O): **δ** = 1.39–1.83 (CH_2_CH(C_4_H_6_NO), br, 2H), 1.84–2.07 (NCH_2_CH_2_CH_2_CO, br, 2H), 2.08–2.50 (NCH_2_CH_2_CH_2_CO, br, 2H), 2.93–3.38 (NCH_2_CH_2_CH_2_CO, br, 2H), 3.41–3.86 (CH_2_CH(C_4_H_6_NO), br 1H).^13^C NMR (500 MHz, D_2_O): **δ** = 17 (NCH_2_CH_2_CH_2_CO), 31(NCH_2_CH_2_CH_2_CO), 35 (CH_2_CH(C_4_H_6_NO), 44 (NCH_2_CH_2_CH_2_CO), 52 (CH_2_CH(C_4_H_6_NO), 163(-NH-CHO), 177 (C=O).IR (KBr) Lactam C=O 1642 cm^−1^.M_n_ SEC (DMF) = 6.900 Da, M_w_/M_n_ = 1.32.Elem. Anal.Found: C 18.58%, H 1.21%, N 4.60%.Calculated: C 18.56%, H 1.25%, N 4.33%.

#### 3.2.2. Synthesis of Copolymer of *N*-Vinylpyrrolidone with *N*-Vinylamine Hydrochloride. Deprotection of PVP–PVFA Copolymers

A weighed amount of *N*-vinylpyrrolidone-*N*-vinylformamide copolymer (0.5 g) was dissolved in 5 mL of 1 M hydrochloric acid, and the solution was boiled for 14 h in a round-bottomed flask equipped with a reflux condenser. Then, the mixture was dialyzed against water for 24 h. After purification, the product was isolated by lyophilization. The content of *N*-vinylamine hydrochloride units in the final copolymer was determined by argentometric titration. The yield was 0.43 g (86%).

^1^H NMR (500 MHz, D_2_O): **δ** = 1.39–1.83 (CH_2_CH(C_4_H_6_NO), br, 2H), 1.84–2.07 (NCH_2_CH_2_CH_2_CO, br, 2H), 2.08–2.50 (NCH_2_CH_2_CH_2_CO, br, 2H), 2.93–3.38 (NCH_2_CH_2_CH_2_CO, br, 2H), 3.41–3.86 (CH_2_CH(C_4_H_6_NO), br 1H).^13^C NMR (500 MHz, D_2_O): **δ** = 17 (NCH_2_CH_2_CH_2_CO), 31(NCH_2_CH_2_CH_2_CO), 35 (CH_2_CH(C_4_H_6_NO)), 44 (NCH_2_CH_2_CH_2_CO), 52 (CH_2_CH(C_4_H_6_NO)), 177 (C=O).IR (KBr) Lactam C=O 1642 cm^−1^.M_n_ SEC (DMF) = 6.900 Da, M_w_/M_n_ = 1.32.Elem. Anal.Found: C 51.15%, H 6.75%, N 11.09%.Calculated: C 51.01%, H 6.78%, N 11.08%. (

#### 3.2.3. Synthesis of Copolymer of *N*-Vinylpyrrolidone with Potassium *N*-Vinyldithiocarbamate

A weighed amount (0.2 g) of *N*-vinylpyrrolidone-*N*-vinylamine hydrochloride copolymer (83:17 mol.%) was dissolved in 5 mL of ethanol, then 66 mg of KOH was added. The mixture was brought to the boil, boiled for 3 min, and cooled down; then, the precipitated potassium chloride was separated by centrifugation. Then, carbon disulfide (0.28 mL) was added to the reaction mixture, and the solution was exposed for 30 min at room temperature and at continuous stirring. In 30 min, the reaction mixture was poured into 100 mL of ethyl acetate/acetone mixture (70/30). The white flaky precipitate was filtered off using a Schott glass filter (pore diameter 40 μm), then reprecipitated twice from ethanol solution into ethyl acetate, and dried under vacuum at 30 °C. The content of potassium *N*-vinyldithiocarbamate units was determined by elemental analysis (sulfur mass portion). The yield of the final product was 0.14 g (70%).

^1^H NMR (500 MHz, D_2_O): **δ** = 1.39–1.83 (CH_2_CH(C_4_H_6_NO), br, 2H), 1.84–2.07 (NCH_2_CH_2_CH_2_CO, br, 2H), 2.08–2.50 (NCH_2_CH_2_CH_2_CO, br, 2H), 2.93–3.38 (NCH_2_CH_2_CH_2_CO, br, 2H), 3.41–3.86 (CH_2_CH(C_4_H_6_NO), br 1H).^13^C NMR (500 MHz, D_2_O): **δ** = 17 (NCH_2_CH_2_CH_2_CO), 31(NCH_2_CH_2_CH_2_CO), 35 (CH_2_CH(C_4_H_6_NO)), 44 (NCH_2_CH_2_CH_2_CO), 52 (CH_2_CH(C_4_H_6_NO)), 177 (C=O).IR (KBr). Lactam C=O 1642 cm^−1^M_n_ SEC (DMF) = 6.900 Da, M_w_/M_n_ = 1.32.Elem. Anal.Found: C 48.66%, H 6.05%, N 10.49%, S 8.08%.Calculated: C 48.50%, H 6.00%, N 10.30%, S 8.01%.

#### 3.2.4. Synthesis of Metal–Polymer Complex between Indium and *N*-Vinylpyrrolidone-Potassium *N*-vinyldithiocarbamate Copolymer

*N*-vinylpyrrolidone-potassium *N*-vinyldithiocarbamate copolymer (0.14 g) was dissolved in 50 mL of water. Then, 3 mL of 0.01 M solution of indium trichloride (with previously added 100 mg of sodium acetate) was added to the obtained solution. The mixture was exposed at room temperature and at stirring for 10 min. The product was purified by dialysis (pore size 1 kDa) against distilled water for 24 h. After purification, the reaction mixture was freeze-dried. The yield of the metal–polymer complex was 122 mg (87.1%).

The yield of the final product was 0.14 g (70%).

^1^H NMR (500 MHz, D_2_O): **δ** = 1.39–1.83 (CH_2_CH(C_4_H_6_NO), br, 2H), 1.84–2.07 (NCH_2_CH_2_CH_2_CO, br, 2H), 2.08–2.50 (NCH_2_CH_2_CH_2_CO, br, 2H), 2.93–3.38 (NCH_2_CH_2_CH_2_CO, br, 2H), 3.41–3.86 (CH_2_CH(C_4_H_6_NO), br 1H).^13^C NMR (500 MHz, D_2_O): **δ** = 17 (NCH_2_CH_2_CH_2_CO), 31(NCH_2_CH_2_CH_2_CO), 35 (CH_2_CH(C_4_H_6_NO)), 44 (NCH_2_CH_2_CH_2_CO), 52 (CH_2_CH(C_4_H_6_NO)), 177 (C=O).IR (KBr). Lactam 1642 cm^−1^ (νC=O), 976 cm^−1^ (νC=S), 1126, 1088, 1051(ν N-C=S)M_n_ SEC (DMF) = 6.900 Da, M_w_/M_n_ = 1.32Elem. Anal.Found: C 48.66%, H 6.05%, N 10.49%, S 8.08%.Calculated: C 48.50%, H 6.00%, N 10.30%, S 8.01%3.

#### 3.2.5. Synthesis of PotassiumBis(Carboxymethyl)Dithiocarbamate

Weighed amounts of iminodiacetic acid (1.33 g, 0.01 mol) and potassium hydroxide (1.68 g, 0.03 mol) were dissolved in 10 mL of water; then, 10 mL of ethanol and 1.2 mL (0.02 mol) of carbon disulfide were added at room temperature and stirred for 1 h. Then, the reaction mixture was left to stand for 2 days; at the end of this period, red oil was formed at the bottom of the flask. This oil was decanted and 30 mL of acetone was added. After grinding with a glass spatula, the oil became thick, then yellowish-white crystals were formed. The precipitated crystals were filtered off and washed twice with 10 mL of acetone. The yield was 1.83 g (56.65%).

^1^H NMR (500 MHz, D_2_O), δ (ppm): 4.62 (2H, s)^13^C NMR (500 MHz, D_2_O), δ (ppm): 211.71 (CSS), 176.54 (−C=O), 171.82(−C=O), 59.34 (CH_2_), 49.22 (CH_2_).Elem. Anal.Found: C 18.58%, H 1.21%, N 4.60%.Calculated: C 18.56%, H 1.25%, N 4.33%.

### 3.3. Instrument and Measurements

H and ^13^C NMR spectra were recorded using a Bruker Avance II-500 WB spectrometer in deuterated solvents (D_2_O) obtained from Sigma-Aldrich. Chemical shifts were measured against residual non-deuterated solvent (water, 4.8 ppm) and external tetramethylsilane standatd.

IR spectra were measured using a Shimadzu Prestige FTIR spectrometer (in KBr pellets).

UV spectra were registered using a Shimadzu UV-1280 spectrophotometer.

Elemental analysis (C, H, N, and S) was performed using a Vario EL-III elemental analyzer. The percentages of carbon, hydrogen, nitrogen, and sulfur were estimated.

Contents of In were determined using an ICP Atomic Emission Spectrometer ICPE-9820 (Shimadzu).

Chromatographic analysis was performed with the use of a Smartline HPLC instrument (KNAUER, Geretsried, Germany) equipped with a Jet Stream column thermostat, and refractometric and spectrophotometric detectors (K-2501 diode array detector, λ = 200–500 nm). Registration of chromatograms and calculations of molecular masses and other parameters were performed using Clarity Chrom GPC/SEC software (Geretsried, Germany). An ultrahydrogel linear SEC column (7.8 × 300 mm)) with a pre-column (0.6 × 40 mm, Waters, Milford, MS, USA) was used for analysis of the copolymers. Analyses were carried out in aqueous solution of 0.2 M NaCl as an eluent at 25 °C. Calibration dependences for the columns were plotted using the data for previously characterized poly(*N*-vinylformamide) standards in 0.2 M aqueous solution of NaCl; the values of the Kuhn–Mark–Houwink constants were K = 10.74 × 10^−3^ and α = 0.76 ± 0.04 [62].

Intrinsic viscosity [η] was measured using an Ubbelohde viscometer. Relative viscosity [η_r_] was calculated as an initial slope of the ln(η_r_) = f(c) dependence, i.e., in the region where η_r_ is the relative viscosity of a solution at concentration *c*. The measurements were performed in 0.1 M solution of sodium acetate at 25 °C.

*Translational diffusion coefficients D* were determined at 24 °C; the technique involved the recording of dispersion of the solution–solvent boundary using the Tsvetkov polarizing diffusometer [60,61]. The images of interference fringes of the solution–solvent boundary were processed using the maximum ordinate method and the method involving measuring areas under interference fringes [62].

Sedimentation coefficients *s* of macromolecules were measured at 24 °C using a MOM 3180 ultracentrifuge (Hungary) equipped with a polarization interferometer attachment [63], at a rotation speed of 40 × 10^3^ rpm. During measurements of diffusion and sedimentation coefficients, concentrations of the solutions did not exceed 0.15 g/dL.

The results of sedimentation and diffusion experiments were used in determining molecular masses of copolymers by the Svedberg method, according to the relationship *M*_sD_ = (*s/D*) × *N*_A_*kT*/(1 − $\bar{v}$*ρ*_0_), where *k* is the Boltzmann constant and *T* is the absolute temperature. The partial specific volume of copolymer ($\bar{v}$) was calculated additively using mass densities ($\bar{v}$^−1^) of the components (poly(*N*-vinylpyrrolidone) and poly(*N*-vinylformamide) [62]), taking into account copolymer composition. The value for the copolymer was found to be v = 0.775 cm^3^/g.

*Hydrodynamic radii* (*R*_h_) were determined by dynamic light scattering (DLS) with the help of a Photocor Complex correlation spectrometer (light source: a coherent He/Ne laser, power output 20 mW, wavelength λ = 632.8 nm) equipped with a Photo or FC programmable correlator (288 channels, ZAO “Anteks”, Russia). The correlation function was processed using DynaLS software (“Gelios”, Russia). This software program allows for the calculating of equivalent sphere hydrodynamic radius *R*_h_ on the basis of the measured diffusion coefficients *D*, according to the Einstein–Stokes equation: *R*_h_ = *kT*/6π*Dη*_s_, where *η*_s_ is the solvent viscosity and *T* is the temperature.

## 4. Conclusions

To summarize, water-soluble chain-flexible VP copolymers with bidentate DTC chelation units were firstly synthesized and characterized. Copolymers with a low content (6 mol.%) of DTC units appeared to be stable in aqueous solutions at pH close to neutral for a long time. In order to evaluate the coordination behavior of group III metals with DTC ligands in aqueous solutions, an IDADTC chelator was synthesized. Stepwise complexation of gallium and indium with this DTC ligand was studied by spectral methods and the formation of a complex with M:L 1:3 ratio was observed. MPC of indium with VP–VDTC were synthesized in aqueous solution, isolated, and characterized. Analogous gallium MPC decompose upon isolation from solution. Spectral data allow one to assume a hexadentate coordination of a DTC-containing polymeric chelation system to metal ion to form intramolecular MPC. Using hydrodynamic and optics methods, decreases in hydrodynamic radius of MPC were estimated in contrast to starting copolymers, while molecular masses were approximately the same. This also indicates the formation of intramolecular MPC. HCR experiments demonstrated that indium MPC were stable in media close to biological in contrast to gallium, which underwent trans-chelation and decomposed to the initial copolymer and low molecular weight complex Ga(His)_2_^+^. Thus, synthesized DTC-containing copolymers of VP seem to be promising macromolecular carriers for III group metal radionuclides for the further development of novel radiopharmaceuticals.

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
