# Peer review of "Synthesis of Water-Soluble Copolymers of N-vinylpyrrolidone with N-vinyldithiocarbamate as Multidentate Polymeric Chelation Systems and Their Complexes with Indium and Gallium"

_molecules, 2020, doi:10.3390/molecules25204681_

Round 1

Reviewer 1 Report

This paper describes syntheses of copolymers of N-vinylpyrrolidone with N-vinyldithiocarbamate and their complexes with gallium and indium. Unfortunately, this study is incomplete because the syntheses and identifications of compounds are described in most of this paper. Thus, this paper should be rejected at present.

Some additional comments are listed below.

1) line 107: What is VP-VA-VDTC. Authors should explain for it.

2) line 194: The M:L ratio should be 1:11, if the given concentration of metal ion is correct.

Author Response

1) line 107: What is VP-VA-VDTC. Authors should explain for it.

It's misprint. Correctly VP-VDTC

2) line 194: The M:L ratio should be 1:11, if the given concentration of metal ion is correct.

It's misprint. Correct concentration CM3+ = 5.83⋅10−3

This paper describes syntheses of copolymers of N-vinylpyrrolidone with N-vinyldithiocarbamate and their complexes with gallium and indium. Unfortunately, this study is incomplete because the syntheses and identifications of compounds are described in most of this paper. Thus, this paper should be rejected at present.

Probably, it was not clear evident from previous version that presented results are novel and original, because data on preparation of water soluble DTC containing polymers and corresponding MPC are scarce, especially for possible biological application.

We engaged such methods as viscosimetry and dynamic light scattering mainly used in polymer chemistry for exact determination of molecular masses and hydrodynamic radius, which are urgent parameters for application of polymers as carriers of drugs or radionuclides. We also applied histidine challenge test, which are widely applied in radiopharmaceutical chemistry for evaluation in vitro stability of radiotracers in non-carrier added level and rarely used for weight amounts.

Isolated MPC are original due to realization of intramolecular structure, where all coordination positions of metal ion are occupied with donor atoms of ligand system. This point is urgent for biological application, because prevent linking of metal coordination compounds by blood proteins.

Thus, we used practically all possible experimental methods used in coordination chemistry, polymer chemistry, radiochemistry for characterization of such complicated objects, like MPC.

Reviewer 2 Report

After reading the manuscript several times, it's difficult for me to gather any actual importance of the findings presented by the authors. The manuscript is interesting but not novel. The quality of figures and results presentation is moderate and need formal as well as qualitative polishing.

As for corrections:

  1. the manuscript should be read carefully and revised, including English language.
  2. The scheme showing complexation reaction should be added. What is the expected coordination number and geometry of the metal ions in formed complexes?
  3. Authors said about shifts in 13C and 1H spectra during complexation reaction. The titration experiment showing the shift of signals should be added along with the corresponding NMR spectra.
  4. In the experimental part, authors said that “N-vinylpyrrolidone (2 g, 18 mmol), N-vinylformamide (0.32 g, 4.5 mmol) and AIBN (azobisisobutyronitrile) initiator (0.07 g, 0.43 mmol, 3 wt.% with respect to total monomer mass) were introduced into heat-resistant glass ampoule. Isopropanol (55.6 ml) was added to the mixture.” As it is known that the AIBN-promoted polymerization is a radical reaction and it is highly sensitive to the presence of oxygen the degassing procedure should be added.
  5. “The sealed ampoule was thermostated at 65ºC for 24 hrs.” but was the solution stirred? It should be
  6. Line 384 1 in 1H is missing.
  7. Chemical shift of the solvent should be added in the NMR procedure.
  8. Some references about synthesis and applications of polymeric complexes should be added (RSC Adv. 2014, 4, 19053-19060, J.  Phys. Chem. 2012, 8, 925-934, Polyhedron, 2006, 25, 2643-2649)
  9. The conclusion of the article should be rewritten to show significance of the obtained results.

Author Response

1. the manuscript should be read carefully and revised, including English language.

Responce1. Hopefully, corrections are introduced

2. The scheme showing complexation reaction should be added. What is the expected coordination number and geometry of the metal ions in formed complexes?

Responce2. We inserted expected structure of MPC complex with comments (lines 380-386)

3. Authors said about shifts in 13C and 1H spectra during complexation reaction. The titration experiment showing the shift of signals should be added along with the corresponding NMR spectra.

Responce3.

Corresponding illustration was inserted (lines 271-274)

4. In the experimental part, authors said that “N-vinylpyrrolidone (2 g, 18 mmol), N-vinylformamide (0.32 g, 4.5 mmol) and AIBN (azobisisobutyronitrile) initiator (0.07 g, 0.43 mmol, 3 wt.% with respect to total monomer mass) were introduced into heat-resistant glass ampoule. Isopropanol (55.6 ml) was added to the mixture.” As it is known that the AIBN-promoted polymerization is a radical reaction and it is highly sensitive to the presence of oxygen the degassing procedure should be added.

Responce 4. Of course, this reaction is air sensitive. Nitrogen purging detail was missed. (line 441)  

5. “The sealed ampoule was thermostated at 65ºC for 24 hrs.” but was the solution stirred? It should be

Responce 5. It is difficult to stir sealed ampoule reaction mixture placed in thermostat.  Thus, reaction mixture just heated. This procedure is well developed and the yields are 80-90%, depending on ratio of monomers  

6. Line 384 1 in 1H is missing.

Responce 6.

1H NMR (500 MHz, D2O), δ (ppm): 4.62 (2H, s) (line 522)

7. Chemical shift of the solvent should be added in the NMR procedure.

Responce 7.

line 531

8. Some references about synthesis and applications of polymeric complexes should be added (RSC Adv. 2014, 4, 19053-19060, J.  Phys. Chem. 2012, 8, 925-934, Polyhedron, 2006, 25, 2643-2649)

Responce 8.

We revised reference list and added more relevant, including RSC Adv. 2014, 4, 19053-1906

9. The conclusion of the article should be rewritten to show significance of the obtained results.

Responce 9.

See lines (575-589)

Reviewer 3 Report

I believe that this work, while not presenting particular errors, is of little interest to readers and of low quality to Molecules.

Moreover:
Figure 1, the lines are too similar and do not stand out well from each other;
Figure 3, better rationalize the trend shown
Page 10 lines 267-272, the considerations are inverted with respect to the figures shown

Author Response

  1. Figure 1, the lines are too similar and do not stand out well from each other;

Responce1.

See line 200

2. Figure 3, better rationalize the trend shown

Responce2.

We did not understood this comment. For our point of view the trend is clear and hardly possible to correct experimental data.

3. Page 10 lines 267-272, the considerations are inverted with respect to the figures shown

Responce 3.

Presented figs. shows, that indium MPC was stable in standard conditions, and gallium MPC underwent trans-chelation and decomposed to the initial copolymer and low molecular weight complex Ga(His)2+

Probably, it was not clear evident from previous version that presented results are novel and original, because data on preparation of water soluble DTC containing polymers and corresponding MPC are scarce, especially for possible biological application. As an example we used publication “Chitosan-Based Bio-Composite Modified withThiocarbamate Moiety for Decontamination of Cations from the Aqueous Media” by Nisar Ali, Adnan Khan, Muhammad Bilal, Sumeet Malik, Syed Badshah, Hafiz M. N. Iqbal, published in Molecules 2020, 25, 226; doi:10.3390/molecules25010226 , because the objects and experimental methods are close. We believe, that the experimental level of our work is not lower, moreover we engaged such methods as viscosimetry and dynamic light scattering mainly used in polymer chemistry for exact determination of molecular masses and hydrodynamic radius, which are urgent parameters for application of polymers as carriers of drugs or radionuclides. We also applied histidine challenge test, which are widely applied in radiopharmaceutical chemistry for evaluation in vitro stability of radiotracers in non-carrier added level and rarely used for weight amounts.

Isolated MPC are original due to realization of intramolecular structure, where all coordination positions of metal ion are occupied with donor atoms of ligand system. This point is urgent for biological application, because prevent linking of metal coordination compounds by blood proteins.

Thus, we used practically all possible experimental methods used in coordination chemistry, polymer chemistry, radiochemistry for characterization of such complicated objects, like MPC.

However, opinion of reviewers about possibility of publication of our materials is absolute and we are not able to contest it.  

Round 2

Reviewer 1 Report

This paper describes syntheses of copolymers of N-vinylpyrrolidone with N-vinyldithiocarbamate and their complexes with gallium and indium. This is carefully done study and the findings are interesting. Thus, this paper is worth publishing in Molecules as present form.